# REGULARIZATION COCKTAILS FOR TABULAR DATASETS

## ABSTRACT

The regularization of prediction models is arguably the most crucial ingredient that allows Machine Learning solutions to generalize well on unseen data. Several types of regularization are popular in the Deep Learning community (e.g., weight decay, drop-out, early stopping, etc.), but so far these are selected on an ad-hoc basis, and there is no systematic study as to how different regularizers should be combined into the best "cocktail". In this paper, we fill this gap, by considering the cocktails of 13 different regularization methods and framing the question of how to best combine them as a standard hyperparameter optimization problem. We perform a large-scale empirical study on 40 tabular datasets, concluding that, firstly, regularization cocktails substantially outperform individual regularization methods, even if the hyperparameters of the latter are carefully tuned; secondly, the optimal regularization cocktail depends on the dataset; and thirdly, regularization cocktails yield the state-of-the-art in classifying tabular datasets by outperforming Gradient-Boosted Decision Trees.

## 1 INTRODUCTION

In most supervised learning application domains, the available data for training predictive models is both limited and noisy with respect to the target variable. Therefore, it is paramount to regularize machine learning models for generalizing the predictive performance on future unseen data. The concept of regularization is well-studied and constitutes one of the pillar components of machine learning. Throughout this work we use the term "regularization" for all methods that explicitly or implicitly take measures that reduce the overfitting phenomenon; we categorize these non-exhaustively into weight decay, data augmentation, model averaging, structure and linearization, and implicit regularization families (detailed in Section 2). In this paper, we propose a new principled strategy that highlights the need for automatically learning the optimal combination of regularizers, denoted as *regularization cocktails*, via a hyperparameter optimization procedure.

Truth be told, combining regularization methods is of course far from being a novel practice per se. As a matter of fact, most modern deep learning models use combinations of a number of regularizers. For instance, EfficientNet (Tan & Q.Le, 2019) mixes components of structural regularization and linearization via ResNet-style skip connections (He et al., 2016), learning rate scheduling, Drop-Out ensembling (Srivastava et al., 2014) and AutoAugment data augmentation (Cubuk et al., 2019). However, even though each of those regularizers is motivated in isolation, the reasoning behind a specific combination of regularizers is largely based on accuracy-driven manual trial-and-error iterations, mostly on image classification benchmarks like CIFAR (Krizhevsky et al., 2009) and ImageNet (Deng et al., 2009).

Unfortunately, the manual search for combinations of regularizers is sub-optimal, unsustainable, and in essence consists of an example of manual hyperparameter tuning, which in turn is easily outperformed by automated algorithms (Snoek et al., 2012; Thornton et al., 2013; Feurer et al., 2015; Olson & Moore, 2016; Jin et al., 2019; Erickson et al., 2020; Zimmer et al., 2020).

Following the spirit of AutoML (Hutter et al., 2018), we, therefore, propose a strategy for learning the optimal dataset-specific regularization cocktail by means of a modern hyperparameter optimization (HPO) method. To the best of our knowledge, there exists no study providing empirical evidence that a mixture of numerous regularizers outperforms individual regularizers; this paper fills this gap. More precisely, the research hypothesis of this paper is that a properly mixed regularization cocktail

outperforms every individual regularizer in it, in terms of accuracy under the same run-time budget, and that the best cocktail to use depends on the dataset. To validate this hypothesis, we executed a large-scale experimental study employing 40 diverse tabular datasets and 13 prominent regularizers with a thorough hyperparameter tuning for all regularizers. We focus on tabular datasets because, in contrast to large image datasets, a thorough hyper-parameter search procedure is feasible. Neural networks are high variance models for tabular datasets, therefore improved regularization schemes can provide a relatively higher generalization gain on tabular datasets compared to other data types.

Thereby, we make the followings contributions:

1. We demonstrate the empirical accuracy gains of regularization cocktails in a systematic manner via a large-scale experimental study on tabular datasets;

2. We challenge the status-quo practices of designing universal dataset-agnostic regularizers, by showing that an optimal regularization cocktail is highly dataset-dependent;

3. We demonstrate that regularization cocktails achieve state-of-the-art classification accuracy performance on tabular datasets and outperform Gradient-Boosted Decision Trees (GBDT) with a statistically-significant margin;

4. As an overarching contribution, this paper provides previously-lacking in-depth empirical evidence to better understand the importance of combining different mechanisms for regularization, one of the most fundamental concepts in machine learning.

## 2 RELATED WORK

**Weight decay:** The classical approaches of regularization focused on minimizing the norms of the parameter values, concretely either the L1 (Tibshirani, 1996), the L2 (Tikhonov, 1943), or a combination of L1 and L2 known as the Elastic Net (Zou & Hastie, 2005). A recent work fixes the malpractice of adding the decay penalty term before momentum-based adaptive learning rate steps (e.g., in common implementations of Adam (Kingma & Ba, 2015)), by decoupling the regularization from the loss and applying it after the learning rate computation (Loshchilov & Hutter, 2019).

**Data Augmentation:** A different treatment of the overfitting phenomenon relies on enriching the training dataset via instance augmentation. The literature on data augmentation is vast, especially for image data, ranging from basic image manipulations (e.g., geometric transformations, or mixing images) up to parametric augmentation strategies such as adversarial and controller-based methods (Shorten & Khoshgoftaar, 2019). For example, Cut-Out (Devries & Taylor, 2017) proposes to mask a subset of input features (e.g., pixel patches for images) for ensuring that the predictions remain invariant to distortions in the input space. Along similar lines, Mix-Up (Zhang et al., 2018) generates new instances as a linear span of pairs of training examples, while Cut-Mix (Yun et al., 2019) suggests super-positions of instance pairs with mutually-exclusive pixel masks. A recent technique, called Aug-Mix (Hendrycks et al., 2020), generates instances by sampling chains of augmentation operations. On the other hand, the direction of reinforcement learning (RL) for augmentation policies was elaborated by Auto-Augment (Cubuk et al., 2019), followed by a technique that speeds up the training of the RL policy (S.Lim et al., 2019). Last but not least, adversarial attack strategies (e.g., FGSM (Goodfellow et al., 2015)) generate synthetic examples with minimal perturbations, which are employed in training robust models (Madry et al., 2018).

**Model Averaging:** Ensembled machine learning models have been shown to reduce variance and act as regularizers (Polikar, 2012). A popular ensemble neural network with shared weights among its base models is Drop-Out (Srivastava et al., 2014), which was extended to a variational version with a Gaussian posterior of the model parameters (Kingma et al., 2015). A follow-up work that is known as Mix-Out (Lee et al., 2020) extends Drop-Out by statistically fusing the parameters of two base models. Furthermore, ensembles can be created using models from the local optima discovered along a single convergence procedure (Huang et al., 2016).

**Structural and Linearization:** One strategy of regularizing deep learning models is to discover dedicated neural *structures* that generalize on particular tasks, such as image classification or Natural Language Processing (NLP). In that context, ResNet adds skip connections across layers (He et al., 2016), while the Inception model computes latent representations by aggregating diverse convolutional filter sizes (Szegedy et al., 2017). The attention mechanism gave rise to the popular

Transformer architecture in the realm of NLP (Vaswani et al., 2017). Recently, EfficientNet is an architecture that easily scales deep convolutional neural networks by controlling only a few hyper-parameters (Tan & Q.Le, 2019).

Besides the aforementioned manually-designed architectures, the stream of Neural Architecture Search (Elsken et al., 2019) focuses on exploring neural connectivity graphs for finding the optimal architectures via reinforcement learning (Zoph & Le, 2017), black-box search (Real et al., 2019) or differentiable solvers (Liu et al., 2019). A recent trend adds a dosage of *linearization* to deep models, where skip connections transfer embeddings from previous less non-linear layers (He et al., 2016; Huang et al., 2017). Along similar lines, the Shake-Shake regularization deploys skip connections in parallel convolutional blocks and aggregates the parallel representations through affine combinations (Gastaldi, 2017), while Shake-Drop extends this mechanism to a larger number of CNN architectures (Yamada et al., 2018).

**Implicit:** The last family of regularizers broadly encapsulates methods which do not directly propose novel regularization techniques but have an *implicit* regularization effect as a virtue of their 'modus operandi' (Arora et al., 2019). For instance, Batch Normalization improves generalization by reducing the internal covariate shifts (Ioffe & Szegedy, 2015), while early stopping of the optimization procedure also yields a similar generalization effect (Yao et al., 2007). On the other hand, stabilizing the convergence of the training routine is another implicit regularization, for instance by introducing learning rate scheduling schemes (Loshchilov & Hutter, 2017). The recent strategy of stochastic weight averaging relies on averaging parameter values from the local optima encountered along the sequence of optimization steps (Izmailov et al., 2018), while another approach conducts updates in the direction of a few 'lookahead' steps (Zhang et al., 2019).

**Positioning in the realm of AutoML:** In contrast to the prior literature, we do not propose a new individual regularization method, but empirically identify the superiority of learning regularization cocktails among a set of existing regularizers from the aforementioned categories. We train dataset-specific cocktails as a hyperparameter optimization (HPO) task (Feurer & Hutter, 2019). In that regard, our work is positioned in the realm of AutoML and is a special case of a combined algorithm selection and hyperparameter optimization (Thornton et al., 2013). We learn the regularization cocktails and optimize the joint hyperparameter configuration space by means of BOHB (Falkner et al., 2018b), which is a variation of Hyperband (Li et al., 2017) with model-based surrogates and is one of the current state of the art approaches for efficient HPO.

## 3 MIXING THE REGULARIZATION COCKTAIL

### 3.1 THE RECIPE AS PROBLEM STATEMENT

A training set is composed of features $\mathbf{X}^{(\text{Train})}$ and targets $\mathbf{y}^{(\text{Train})}$, while the test dataset is denoted by $\mathbf{X}^{(\text{Test})}, \mathbf{y}^{(\text{Test})}$. A parameterized function approximates the targets as $\hat{\mathbf{y}} = f(\mathbf{X}; \boldsymbol{\theta})$ whereas the parameters $\boldsymbol{\theta}$ are trained to minimize a differentiable loss function $\mathcal{L}$ as:

$$\boldsymbol{\theta}^* \in \arg\min_{\boldsymbol{\theta}} \ \mathcal{L}\left(\mathbf{y}^{(\text{Train})}, f\left(\mathbf{X}^{(\text{Train})}; \boldsymbol{\theta}\right)\right). \tag{1}$$

To generalize into minimizing $\mathcal{L}\left(\mathbf{y}^{(\text{Test})}, f(\mathbf{X}^{(\text{Test})}; \boldsymbol{\theta}\right)$, the parameters of $f$ are controlled with a regularization technique $\Omega$ that avoids overfitting to the peculiarities of the training data. With a slight abuse of notation we denote $f\left(\mathbf{X}; \Omega\left(\boldsymbol{\theta}; \boldsymbol{\lambda}\right)\right)$ to be the predictions of the model $f$ whose parameters $\boldsymbol{\theta}$ are optimized under the regime of the regularization method $\Omega(\cdot; \boldsymbol{\lambda})$, where $\boldsymbol{\lambda} \in \boldsymbol{\Lambda}$ represent the hyperparameters of $\Omega$. The training data is further divided into two subsets as training and validation splits,[1] the later denoted by $\mathbf{X}^{(\text{Val})}, \mathbf{y}^{(\text{Val})}$, such that $\boldsymbol{\lambda}$ can be tuned on the validation loss via the following hyperparameter optimization objective:

$$\boldsymbol{\lambda}^* \quad \in \quad \arg\min_{\boldsymbol{\lambda} \in \boldsymbol{\Lambda}} \ \mathcal{L}\left(\mathbf{y}^{(\text{Val})}, f\left(\mathbf{X}^{(\text{Val})}; \Omega\left(\boldsymbol{\theta}^*; \boldsymbol{\lambda}\right)\right)\right), \tag{2}$$

$$\text{s.t.} \quad \boldsymbol{\theta}^* \in \arg\min_{\boldsymbol{\theta}} \ \mathcal{L}\left(\mathbf{y}^{(\text{Train})}, f(\mathbf{X}^{(\text{Train})}; \Omega\left(\boldsymbol{\theta}; \boldsymbol{\lambda}\right))\right). \tag{3}$$

---

[1]For simplicity, we only discuss hold-out validation scheme here, but in principle any other validation scheme, such as cross validation and bootstrap sampling, would be possible.

While the search for optimal hyperparameters $\boldsymbol{\lambda}$ is an active field of research in the realm of AutoML (Hutter et al., 2018), still the choice of the regularizer $\Omega$ mostly remains an ad-hoc practice, where practitioners select few combinations among popular regularizers (Dropout, L2, Batch Normalization, etc.). In contrast to prior studies, we hypothesize that the optimal regularizer is a cocktail mixture of a large set of regularization methods, all being simultaneously applied with different strengths (i.e., dataset-specific hyperparameters). Given a set of $K$ regularizers $\left\{\left(\Omega^{(k)}\left(\cdot;\boldsymbol{\lambda}^{(k)}\right)\right\}_{k=1}^{K} := \left\{\Omega^{(1)}\left(\cdot;\boldsymbol{\lambda}^{(1)}\right),\ldots,\Omega^{(K)}\left(\cdot;\boldsymbol{\lambda}^{(K)}\right)\right\}$, each with its own hyperparameters $\boldsymbol{\lambda}^{(k)} \in \boldsymbol{\Lambda}^{(k)}, \forall k \in \{1,\ldots,K\}$, the problem of finding the optimal cocktail of regularizers is:

$$\boldsymbol{\lambda}^* \quad \in \quad \underset{\boldsymbol{\lambda}^{(1)} \in \boldsymbol{\Lambda}^{(k)},\ldots,\boldsymbol{\lambda}^{(K)} \in \boldsymbol{\Lambda}^{(k)}}{\arg\min} \mathcal{L}\left(\mathbf{y}^{(\text{Val})}, f\left(\boldsymbol{X}^{(\text{Val})}; \left\{\Omega^{(k)}\left(\boldsymbol{\theta}^*, \boldsymbol{\lambda}^{(k)}\right)\right\}_{k=1}^{K}\right)\right) \quad (4)$$

$$\text{s.t.} \quad \boldsymbol{\theta}^* \in \underset{\boldsymbol{\theta}}{\arg\min} \ \mathcal{L}\left(\mathbf{y}^{(\text{Train})}, f\left(\mathbf{X}^{(\text{Train})}; \left\{\Omega^{(k)}\left(\boldsymbol{\theta}, \boldsymbol{\lambda}^{(k)}\right)\right\}_{k=1}^{K}\right)\right) \quad (5)$$

The intuitive interpretation of Equations 4-5 is searching for the optimal hyperparameters $\boldsymbol{\lambda}$ (i.e., strengths) of the cocktail's regularizers using the validation set (Equation 4), given that the optimal prediction model parameters $\boldsymbol{\theta}$ are trained under the regime of all the regularizers being applied jointly (Equation 5). We stress that the hyperparameters $\boldsymbol{\lambda}^{(k)}$ include a conditional hyperparameter controlling whether the $k$-th regularizer is applied at all, or skipped. Therefore, the best cocktail might consist of combinations of a subset of regularizers.

## 3.2 Regularization Ingredients and the Search Space

To build the regularization cocktails we combine the 13 methods shown in Table 1, which are selected among the categories of regularizers covered in Section 2, each having its own hyperparameter search space. We set the other hyperparameters regarding the architecture and the optimizer as detailed in Table 2 in Appendix B.1. The regularization cocktails introduce 9 non-conditional hyperparameters in the search space, which, in turn, can add up to 9 conditional hyperparameters. In total, our regularization cocktails can add up to 18 hyperparameters in the search space.

In the defined search space some of the combinations are not technically feasible, therefore, we introduce the following constraints to the proposed search space: (i) Shake-Shake and Shake-Drop are not simultaneously active since the latter builds on the former. (ii) Only one data augmentation technique out of Mix-Up, Cut-Mix, Cut-Out, and FGSM adversarial learning can be active at once due to a technical limitation of the base library (Zimmer et al., 2020).

As an optimizer, we decided to use BOHB (Falkner et al., 2018a) since it achieves a strong anytime performance by combining Hyperband (Li et al., 2017) and Bayesian Optimization (Shahriari et al., 2016), and still has the convergence guarantees of Hyperband. Furthermore, BOHB can deal with the categorical hyperparameters for enabling or disabling regularization techniques and the corresponding conditional structures. In Appendix A we provide a brief description of how BOHB works.

## 4 Experimental Protocol

### 4.1 Experimental Setup

We use a collection of tabular datasets (listed in Table 4 in Appendix D) from the recent open-source OpenML AutoML Benchmark (Gijsbers et al., 2019), as well as popular online repositories, such as UCI (Asuncion & Newman, 2007) and Kaggle. Our benchmark of 40 datasets includes tabular datasets that represent diverse classification problems, containing between 452 and 416188 instances, and between 4 and 2001 features, varying in terms of the number of numerical and categorical features. The datasets are retrieved from OpenML (Vanschoren et al., 2014) and split as 60% training, 20% validation and 20% testing sets. All data is standardized to have 0 mean and unit variance. For datasets with missing values, the median value on the known values is used as an imputation strategy.

We ran all experiments on a CPU cluster, each node of which contains two Intel Xeon E5-2630v4 at 2.2GHz with 20 CPU cores and a total memory of 128GB. We chose the PyTorch library (Paszke

| Regularizer | Hyperparameter | Type | Range | Conditionality |
|---|---|---|---|---|
| BN | BN-active | Boolean | {True, False} | − |
| WD | WD-active | Boolean | {True, False} | − |
|  | Decay factor | Continuous | $[10^{-5}, 0.1]$ | WD-active |
| DO | DO-active | Boolean | {True, False} | − |
|  | Dropout shape | Nominal | {funnel, long funnel, diamond, hexagon, brick, triangle, stairs} | DO-active |
|  | Drop rate | Continuous | $[0.0, 0.8]$ | DO-active |
| SC | SC-active | Boolean | {True, False} | − |
|  | MB choice | Nominal | {SS, SD, Standard} | SC-active |
| SD | Max. probability | Continuous | $[0.0, 1.0]$ | SC-active $\wedge$ MB choice = SD |
| LA | LA-active | Boolean | {True, False} | − |
|  | Step size | Continuous | $[0.5, 0.8]$ | LA-active |
|  | Num. steps | Integer | $[1, 5]$ | LA-active |
| SWA | SWA-active | Boolean | {True, False} | - |
| SE | SE-active | Boolean | {True, False} | - |
|  | Num. models | Constant | {3} | SE-active |
| − | Augment | Nominal | {MU, CM, CO, AT, None} | − |
| MU | Mix. magnitude | Continuous | $[0.0, 1.0]$ | Augment = MU |
| CM | Probability | Continuous | $[0.0, 1.0]$ | Augment = CM |
| CO | Probability | Continuous | $[0.0, 1.0]$ | Augment = CO |
|  | Patch ratio | Continuous | $[0.0, 1.0]$ | Augment = CO |

Table 1: The configuration space for the regularization cocktail regarding the **explicit regularization hyperparameters** of the methods and the conditional constraints enabling or disabling them. (BN: Batch Normalization, WD: Weight Decay, DO: Drop-Out, SC: Skip Connection, MB: Multi-branch choice, SS: Shake-Shake, SD: Shake-Drop, LA: Lookahead Optimizer, SWA: Stochastic Weight Averaging, SE: Snapshot Ensembles, MU: Mix-Up, CM: Cut-Mix, CO: Cut-Out, and AT: FGSM Adversarial Learning)

et al., 2019) as the main deep learning framework for our work and we extended the AutoDL-framework Auto-Pytorch (Mendoza et al., 2018; Zimmer et al., 2020) with our implementations for the regularizers, as shown in Table 1.

To optimally utilize resources, we ran BOHB with 10 workers in parallel, where each worker had access to 2 CPU cores and 12GB of memory, executing one configuration at a time. In view of limited computational resources and taking into account the dimensions $D$ of the considered configuration spaces, we ran BOHB for at most 4 days, or at most $40 \times D$ hyperparameter configurations, whichever came first. During the training phase, each configuration was run for 105 epochs. For the sake of studying the effect on more datasets, we only evaluated a single train-val-test split. After the training phase is completed, we report the results on the best hyperparameter configuration found retrained on the joint train and validation set for 105 epochs.

## 4.2 FIXED ARCHITECTURE AND OPTIMIZATION HYPERPARAMETERS

In order to focus exclusively on investigating the effect of individual regularization methods, we fix the hyperparameters that are related to the model architecture and general training procedure in the search-space, as specified in Table 2 of Appendix B.1. These hyperparameter values are tuned for maximizing the performance of an unregularized neural network on our dataset collection (see Table 4 in Appendix D). Moreover, we use a 9-layer feed-forward neural network with 512 units for each layer, a choice based on a previous related work (Orhan & Pitkow, 2017). We emphasize that the network has a sufficiently large capacity, to ensure that the effect of regularization methods would be noticeable.

Moreover, we set a low learning rate of $10^{-3}$ after performing a grid search for finding the value performing best across all datasets. We use the AdamW implementation (Loshchilov & Hutter, 2019), which implements decoupled weight decay, and cosine annealing with restarts (Loshchilov & Hutter, 2019) as a learning rate scheduler. Using a learning rate scheduler with restarts helps in our case because we keep a fixed initial learning rate. For the restarts, we use an initial budget of 15 epochs, with a budget multiplier of 3, following published practices (Zimmer et al., 2020). Additionally, since our benchmark includes imbalanced datasets, we use a weighted version of categorical cross-entropy and balanced accuracy (Brodersen et al., 2010) as the evaluation metric.

### 4.3 HYPOTHESES AND ASSOCIATED EXPERIMENTS

**Hypothesis 1:** The regularization cocktails achieve better generalization performance compared to the individual regularization methods over all datasets.

**Experiment 1:** We regularize the plain neural network (Section 4.2) with each method from Table 1, one at a time. For every regularizer, we tune its hyperparameters on each dataset, then finally measure the regularized network's performance on the test set after retraining the best hyper-parameter configuration on the joint train and validation set. For each dataset, we compare the results against the results of a cocktail optimized the same way as each of the individual ingredient regularizers.

**Hypothesis 2:** The optimal regularization cocktails are dataset-dependent.

**Experiment 2:** We study the best-found regularization cocktails of every dataset and frequencies of the regularizers that were chosen to be activated by BOHB, to demonstrate that no combination of regularizers is frequent. Furthermore, we regularize the plain network with the most frequent regularizers and compare it against our proposed method of Section 3.

**Hypothesis 3:** The regularization cocktails achieve state-of-the-art classification accuracies in tabular datasets.

**Experiment 3:** We compare against GBDT, the state-of-the-art classifier for tabular data. For a fair comparison, we optimized the hyper-parameters of GBDT on every dataset using the popular AutoSklearn[2] library, by following the exact hyper-parameter search protocol (same train, validation, and test splits) and provided GBDT with the same HPO budget as our proposed method. The search space for the hyperparameters of GBDT is further detailed in Section 5.

## 5 EXPERIMENTAL RESULTS

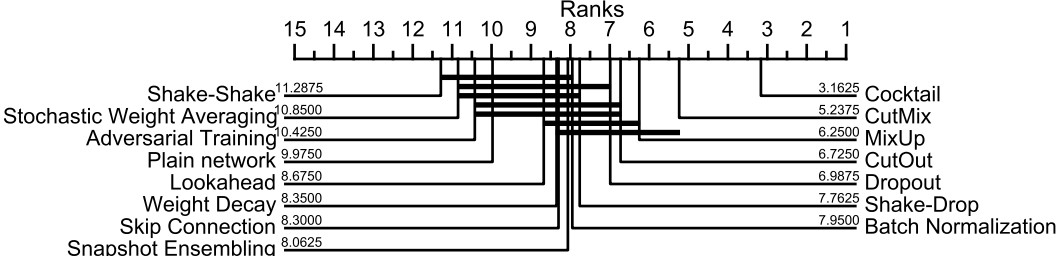

Figure 1: **Critical difference diagram** generated with the Wilcoxon-Holm post-hoc analysis on 40 datasets. The diagram shows the ranks and the statistical significance of the results for every individual regularization technique and our regularization cocktails.

**Regularization cocktail performance (Experiment 1):** Figure 1 presents the critical difference diagram of the ranks, which demonstrates that the cocktail outperforms each individual regularizer. The critical difference diagram is generated by performing a posthoc analysis based on the Wilcoxon-Holm method (Wilcoxon, 1992; Holm, 1979) with a $p$ value of 0.05 as the threshold for

---

[2]https://automl.github.io/auto-sklearn

statistical significance. Observing the results, the regularization cocktail manages to outperform all individual cocktail ingredients with a statistically significant margin. This confirms our hypothesis that well-tuned regularization cocktails outperform well-tuned individual regularization techniques across a diverse suite of tabular datasets. In addition, Figure 2 provides additional information on the rank distributions of the different compared methods, while Figure 6 of Appendix C offers detailed descriptive statistics for each one-on-one comparison against baselines.

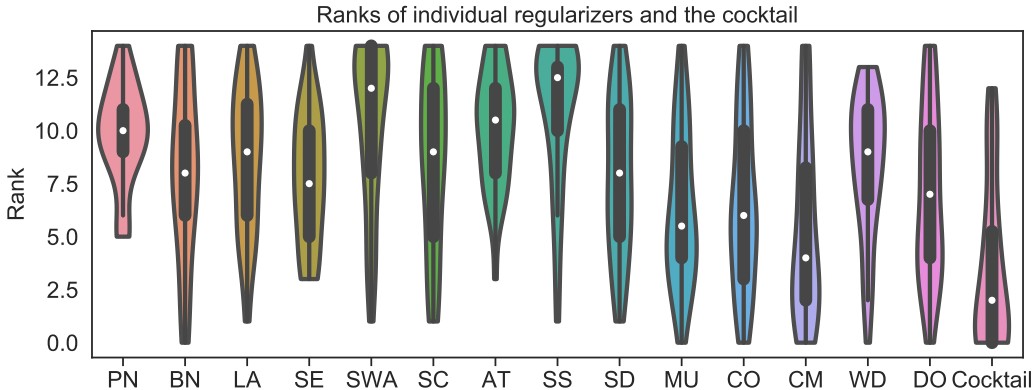

Figure 2: **Rank distribution for the individual regularization methods and the regularization cocktail.** The rank distribution for each method is calculated on the test set over all datasets.

**Dataset-dependent optimal cocktails (Experiment 2):** Figure 3 shows that the optimal regularization cocktail depends on the dataset at hand since no combination of regularizers was active on the majority of the datasets. The plot depicts all frequent singular regularizers and combinations of pairs of regularizers occurring in at least 30% of the datasets, based on how often they were part of the per-dataset cocktails. The most frequent pair of regularizers (BN and SE) is selected only on $50\%$ of the datasets, which highlights the fact that regularization cocktails are dataset-specific and there is no frequent universal combination.

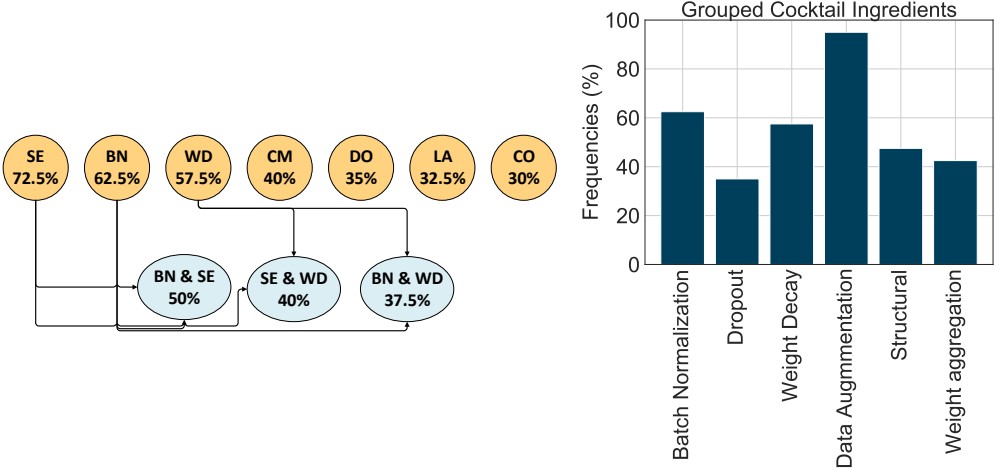

Figure 3: Left: Individual and pairwise cocktail ingredients occurring in at least 30% of the datasets. Right: Clustered histogram of cocktail ingredients. Data Augmentation: {CutMix, Cutout, Mixup, Adversarial Training}, Structural: {Skip connection, Shake-Shake, Shake-Drop}, Weight aggregation: {Lookahead Optimizer, Stochastic Weight Averaging, Snapshot Ensembling}.

Moreover, the results presented in Figure 3 provide insights into frequent cocktail ingredients with regards to the regularization types. For instance, although Snapshot Ensembling as an individual

method ranks 7-th among 13 regularizers in Figure 1, it is nevertheless present in the regularization cocktails of 72.5% datasets. The finding hints that optimal cocktails are composed of weaker regularizers, whose dataset-dependent combination enhances the regularization effect. For a more in-depth summary of the frequencies that correspond to the individual regularization methods, we refer to Figure 7 in Appendix C.

Our Dataset-specific vs. Top-5 Most Frequent (22 wins, 15 losses, 3 draws, $p$-value: 0.08143)

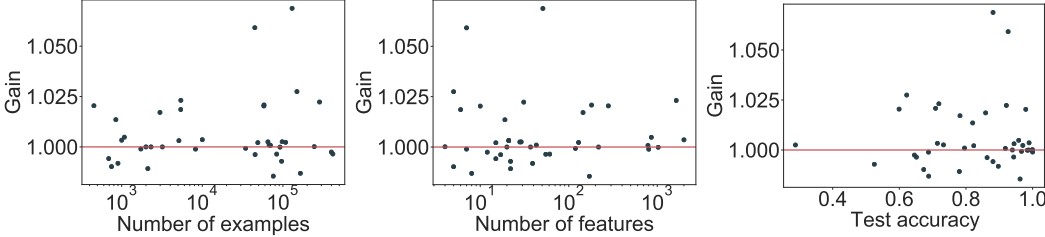

Our Dataset-specific vs. Top-5 Highest Ranks (32 wins, 6 losses, 2 draws, $p$-value: 0.00004)

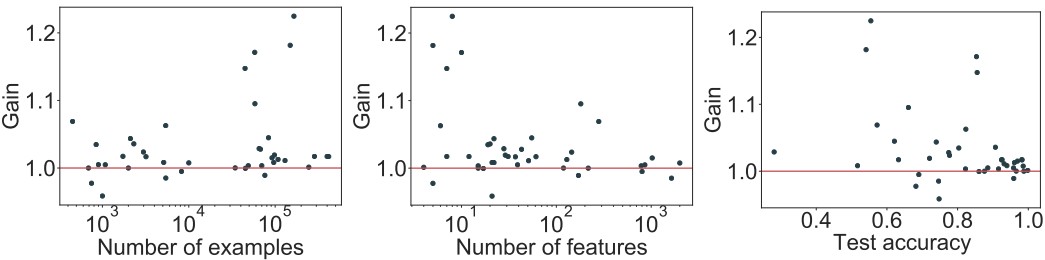

Figure 4: Comparison of our proposed dataset-specific cocktail against the cocktail of the top-5 most frequent regularizers (top row), and the cocktail of the top-5 regularizers with the highest performance ranks (bottom row). Each point represents a dataset and the gain is defined as the test accuracy of our method divided by the test accuracy of each baseline. We illustrate the gain with three ablations: the number of samples (left), number of features (middle) and test accuracy (right).

Lastly, to further validate that good performing regularization cocktails are dataset-dependent, we conducted another experiment by creating 2 baselines consisting of the following top-5 cocktails:

1. The top-5 most frequent regularizers of Experiment 2 (Snapshot Ensembling, Batch Normalization, Dropout, Weight Decay, and dataset-specific augmentation);

2. The top-5 regularizers with the highest ranks from Experiment 1 (Dropout, Shake-Drop, Batch Normalization, Snapshot Ensembling, and dataset-specific augmentation).

In both top-5 cocktails "dataset-specific augmentation" signifies having data augmentation activated, however, the choice between CutMix, CutOut, and Mixup is dataset-specific and is tuned during the HPO process. This design decision was taken to make the baselines even more competitive. The aforementioned regularizers in the top-5 baselines are always applied jointly (i.e. no subset of those methods are selected on a per dataset basis), however, we tune the hyper-parameters of all regularizers in each top-5 baseline jointly for each dataset. We observe that both top-5 baselines underperform against our proposed dataset-specific cocktail as indicated in Figure 4. Additionally, we measured the statistical significance between the top-5 baselines and our method using the Wilcoxon signed-rank test at a 10% significance level. For the top-5 highest ranks variant, the result confirms that the difference is significant with a $p$-value of 0.00004. Similarly, the results show a significance against the top-5 most frequent variant with a $p$-value of 0.08143. For a detailed summary of all the results for every method we refer to Appendix D, Table 5.

**Regularization cocktails achieve state-of-the-art classification accuracy in tabular datasets (Experiment 3):** To investigate whether the regularization cocktails achieve state-of-the-art classification accuracy, we compare our method against the GBDT method, which is the de-facto state-of-the-art in tabular datasets. The results, as presented in Figure 5 show the superiority of the

regularization cocktails in terms of the predictive performance under the same time and resource constraints compared to the GBDT algorithm. We used the GBDT implementation of Auto-Sklearn, a popular automated tool in the realm of AutoML. For ensuring a fair comparison we ran GBDT with the same setup (same training, validation, testing splits) and the same hyper-parameter search time. In addition, we ran experiments with the same computational hardware as for the cocktail. More details on the experimental setup are presented in Appendix B.2.

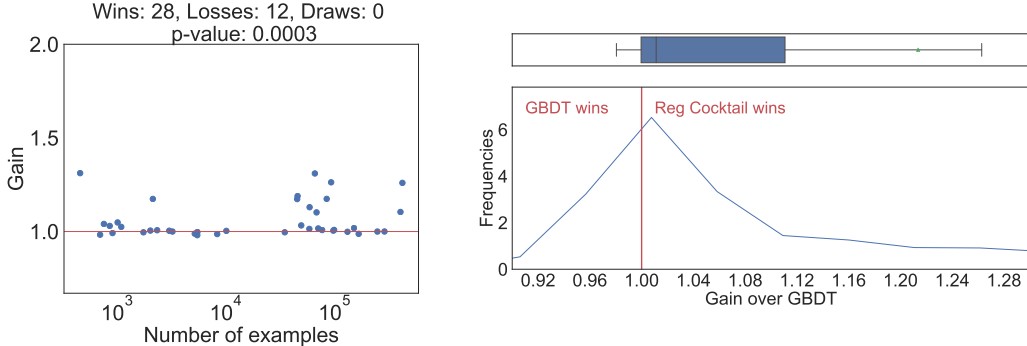

Figure 5: Left: Cocktail gain over the GBDT algorithm calculated on the test set for every dataset (The gain is calculated by dividing the cocktail accuracy with the GBDT accuracy). Right: The distribution of the cocktail gain.

The regularization cocktails achieve higher accuracies compared to GBDT on 28 out of 40 datasets (70% win ratio) and the difference is statistically significant. Figure 5 further illustrates that the performance gain of the regularization cocktails is invariant to the dataset size. In the left subplot of Figure 5 we observe that our method outperforms GBDT in both small and large datasets. Furthermore, the right subplot of Figure 5 depicts the fact that our gain is not marginal and in certain datasets, we achieve up to 30% increase in test accuracy. The full per-dataset accuracies of GBDT are found in Appendix D, Table 7. Lastly, we computed the statistical significance between the cocktail and GBDT using the Wilcoxon signed-rank test, which resulted in a $p$-value of $0.0003$. Based on the empirical results, we conclude that the regularization cocktails yield state-of-the-art prediction models for classifying tabular datasets.

## 6 CONCLUSIONS AND FUTURE WORK

Even though combining regularizers is a relatively frequent practice by researchers, to date, there exists no prior work that systematically studies the effect of **optimally** combining regularization methods. This paper presented the first step in empirically studying regularization cocktails, by posing the problem as a standard hyperparameter optimization challenge. We conducted a large-scale experiment involving 13 regularization methods and 40 datasets, with a thorough hyperparameter optimization procedure for each technique. The findings of this study can be summarized as **three simple take-home messages** for practitioners:

1. Instead of applying a single regularization technique, we recommend exploiting the complementary effects of regularization cocktails.

2. To make neural networks achieve state-of-the-art classification accuracy in classifying tabular datasets the regime of regularization cocktails should be applied.

3. To obtain a well-performing, dataset-specific regularization cocktail, using state-of-the-art hyperparameter optimization techniques is recommended.

As future work, we would like to combine regularization cocktails for neural networks with automated data preprocessing pipelines and architecture search, in order to advance the performance gain of deep learning on small tabular data.

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

## A    DESCRIPTION OF BOHB

---
**Algorithm 1** BOHB
---

   **Input:** $b_{min}$, $b_{max}$ and $\eta$       ▷ b stands for budget, while $\eta$ stands for the downsampling rate
   **Initialization:** $s_{max} = \log_\eta \frac{b_{max}}{b_{min}}$
   **for** $s$ in $(s_{max}, s_{max-1},...,0)$ **do**
      $n = \frac{s_{max}+1}{s+1}$                        ▷ Number of configurations
      Call SH routine with $\eta^{-s} \cdot b_{max}$ as an initial budget    ▷ SH = Successing Halving routine
   **end for**
   **Output:** *Return hyperparameter configuration with the smallest loss*

---

BOHB (Falkner et al., 2018a) is a hyperparameter optimization algorithm that extends Hyperband (Li et al., 2017) by sampling from a model instead of sampling randomly from the hyperparameter search space.

Initially, BOHB performs random search and favors exploration. As it iterates and gets more observations, it builds models over different fidelities and trades off exploration with exploitation to avoid converging in bad regions of the search space. BOHB samples from the model of the highest fidelity with a probability $p$ and with $1 - p$ from random. A model is built for a fidelity only when enough observations exist, by default the criteria is set to equal $S + 1$, where $S$ is the dimensionality of the search space.

BOHB achieves strong anytime results by combining Random Search and Bayesian optimization and helps deep neural networks in achieving faster convergence compared with traditional Bayesian Optimisation methods.

## B    CONFIGURATION SPACES

### B.1    METHOD IMPLICIT SEARCH SPACE

| Category | Hyperparameter | Type | Range | Conditionality |
|---|---|---|---|---|
| Cosine Annealing | Iterations multiplier | Continuous | {2.0} | Scheduler = COS |
| | Max. iterations | Integer | {15} | Scheduler = COS |
| Network | Activation | Nominal | {ReLU} | – |
| | Bias initialization | Nominal | {Yes} | – |
| | Blocks in a group | Integer | {2} | – |
| | Embeddings | Nominal | {None} | – |
| | Number of groups | Integer | {2} | – |
| | Resnet shape | Nominal | {Brick} | Type = Shaped-Resnet |
| | Type | Nominal | {Shaped-Resnet} | – |
| | Units in a layer | Integer | {512} | – |
| Preprocessing | Preprocessor | Nominal | {None} | – |
| Resampling | Target size | Nominal | {Median, Upsample} | |
| | Under sampling | Nominal | {Random, None} | – |
| Training | Batch size | Integer | {128} | – |
| | Imputation | Nominal | {Median} | – |
| | Initialization method | Nominal | {Default} | – |
| | Learning rate | Continuous | {$10^{-3}$} | – |
| | Loss module | Nominal | {Weighted Cross-Entropy} | – |
| | Normalization strategy | Nominal | {Standardize} | – |
| | Optimizer | Nominal | {AdamW} | – |
| | Scheduler | Nominal | {COS} | – |
| | Seed | Integer | {11} | – |

Table 2: The configuration space of the training and model architecture hyperparameters.

Table 2 presents the implicit search space used in all our experiments. The implicit search space is shared between all the individual regularizers and the regularization cocktail.

## B.2 Auto-Sklearn: Gradient Boosted Decision Tree Search Space

For Experiment 3, we set up the search space of Auto-Sklearn as follows:

| Hyperparameter | Type | Range | Conditionality |
|---|---|---|---|
| Early Stopping | Nominal | {Off, Train, Valid} | Estimator = Gradient Boosting |
| $L_2$ Regularization | Continuous | $[1e-10, 1]$ | Estimator = Gradient Boosting |
| Learning Rate | Continuous | $[0.01, 1]$ | Estimator = Gradient Boosting |
| Max Leaf Nodes | Integer | $[3, 2047]$ | Estimator = Gradient Boosting |
| Min Samples Leaf | Integer | $[1, 200]$ | Estimator = Gradient Boosting |
| # Iterations No Change | Integer | $[1, 20]$ | Estimator = Gradient Boosting |
| Validation Fraction | Continuous | $[0.01, 0.4]$ | Estimator = Gradient Boosting |

Table 3: The search space of the training and model hyperparameters for the gradient boosting estimator of the Auto-Sklearn tool.

Furthermore, the estimator for Auto-Sklearn is restricted to only include GBDT, for the sake of fully comparing against the algorithm as a baseline. We do not activate any preprocessing since also our regularization cocktails do not make use of preprocessing algorithms in the pipeline. The time left is always selected based on the time it took BOHB to find the hyperparameter with the best validation accuracy from the start of the hyperparameter optimization phase. The ensemble size is kept to 1 since our method only features one classifier and not multiple ones. The seed is set to 11 as it was set in the experiments with the regularization cocktail, so we can have the same data splits. To keep the comparison fair, there is no warm start for the initial configurations with meta-learning, since, our method also does not make use of meta-learning. Lastly, the number of workers in parallel is set to 10 to match the parallel resources that were given to the experiment with the regularization cocktails.

## C Plots

### C.1 Experiment 1: Regularization Cocktail Performance

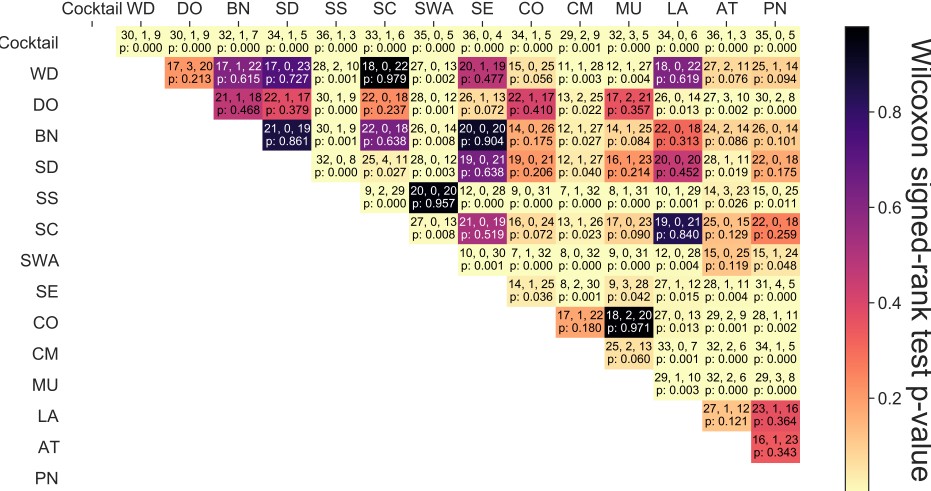

Figure 6: **Pairwise statistical significance and comparison.** For every entry, **the first row** showcases the wins, draws and losses of the horizontal method with the vertical method on all datasets, calculated on the test set; **the second row** presents the p-value for the statistical significance test.

In Figure 6, we present the results of each pairwise comparison. The results presented are calculated on the test set after the refit phase is completed on the best hyperparameter configuration. The p-value is generated by performing the Wilcoxon signed-rank test. As can be seen from the results, the regularization cocktail is the only method that has statistically significant results compared to all the other methods.

## C.2 EXPERIMENT 2: DATASET-DEPENDENT OPTIMAL COCKTAILS

In Figure 7, we present the occurrences of every regularization method over all datasets. The occurrences are calculated by analyzing the best-found hyperparameter configuration for each dataset and observing the number of times the regularization method was chosen to be activated by BOHB.

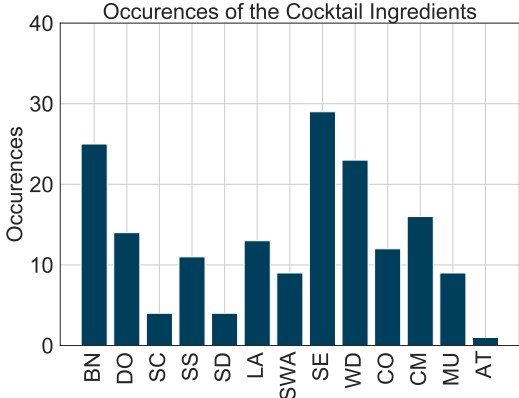

Figure 7: **Frequency of the regularization techniques.** The occurrences of the individual regularization techniques in the best hyperparameter configurations found by the cocktail across 42 datasets.

# D   TABLES

In this section, at Table 4 we provide information about the datasets that are considered in our experiments. Concretely, we provide descriptive statistics and the identifiers for every dataset. The identifier (the task id) can be used to download the datasets from OpenML.

| Task Id | Dataset Name | Number of Instances | Number of Features | Majority Class Percentage | Minority Class Percentage |
|---|---|---|---|---|---|
| 233090 | anneal | 898 | 39 | 76.17 | 0.89 |
| 233091 | kr-vs-kp | 3196 | 37 | 52.22 | 47.78 |
| 233092 | arrhythmia | 452 | 280 | 54.20 | 0.44 |
| 233093 | mfeat-factors | 2000 | 217 | 10.00 | 10.00 |
| 233088 | credit-g | 1000 | 21 | 70.00 | 30.00 |
| 233094 | vehicle | 846 | 19 | 25.77 | 23.52 |
| 233096 | kc1 | 2109 | 22 | 84.54 | 15.46 |
| 233099 | adult | 48842 | 15 | 76.07 | 23.93 |
| 233102 | walking-activity | 149332 | 5 | 14.73 | 0.61 |
| 233103 | phoneme | 5404 | 6 | 70.65 | 29.35 |
| 233104 | skin-segmentation | 245057 | 4 | 79.25 | 20.75 |
| 233106 | ldpa | 164860 | 8 | 33.05 | 0.84 |
| 233107 | nomao | 34465 | 119 | 71.44 | 28.56 |
| 233108 | cnae-9 | 1080 | 857 | 11.11 | 11.11 |
| 233109 | blood-transfusion | 748 | 5 | 76.20 | 23.80 |
| 233110 | bank-marketing | 45211 | 17 | 88.30 | 11.70 |
| 233112 | connect-4 | 67557 | 43 | 65.83 | 9.55 |
| 233113 | shuttle | 58000 | 10 | 78.60 | 0.02 |
| 233114 | higgs | 98050 | 29 | 52.86 | 47.14 |
| 233115 | Australian | 690 | 15 | 55.51 | 44.49 |
| 233116 | car | 1728 | 7 | 70.02 | 3.76 |
| 233117 | segment | 2310 | 20 | 14.29 | 14.29 |
| 233118 | Fashion-MNIST | 70000 | 785 | 10.00 | 10.00 |
| 233119 | Jungle-Chess-2pcs | 44819 | 7 | 51.46 | 9.67 |
| 233120 | numerai28.6 | 96320 | 22 | 50.52 | 49.48 |
| 233121 | Devnagari-Script | 92000 | 1025 | 2.17 | 2.17 |
| 233122 | helena | 65196 | 28 | 6.14 | 0.17 |
| 233123 | jannis | 83733 | 55 | 46.01 | 2.01 |
| 233124 | volkert | 58310 | 181 | 21.96 | 2.33 |
| 233126 | MiniBooNE | 130064 | 51 | 71.94 | 28.06 |
| 233130 | APSFailure | 76000 | 171 | 98.19 | 1.81 |
| 233131 | christine | 5418 | 1637 | 50.00 | 50.00 |
| 233132 | dilbert | 10000 | 2001 | 20.49 | 19.13 |
| 233133 | fabert | 8237 | 801 | 23.39 | 6.09 |
| 233134 | jasmine | 2984 | 145 | 50.00 | 50.00 |
| 233135 | sylvine | 5124 | 21 | 50.00 | 50.00 |
| 233137 | dionis | 416188 | 61 | 0.59 | 0.21 |
| 233142 | aloi | 108000 | 129 | 0.10 | 0.10 |
| 233143 | C.C.FraudD. | 284807 | 31 | 99.83 | 0.17 |
| 233146 | Click prediction | 399482 | 12 | 83.21 | 16.79 |

Table 4: **Datasets.** The collection of datasets used in our experiments, combined with detailed information for each dataset.

Moreover, Table 5 shows the results for the comparison between the Regularization Cocktail and the Top-5 cocktail variants as described in Experiment 2. The results are calculated on the test set for all datasets, after retraining on the best dataset-specific hyperparameter configuration.

| Task Id | Cockt. | Top-5 F | Top-5 R | Task Id | Cockt. | Top-5 F | Top-5 R | Task Id | Cockt. | Top-5 F | Top-5 R |
|---|---|---|---|---|---|---|---|---|---|---|---|
| 233090 | 89.27 | 89.71 | 88.54 | 233091 | 99.85 | 99.85 | 98.20 | 233092 | 61.46 | 59.94 | 57.21 |
| 233093 | 98.00 | 98.75 | 98.75 | 233088 | 74.64 | 71.43 | 74.76 | 233094 | 82.58 | 82.01 | 80.33 |
| 233096 | 74.38 | 78.03 | 73.96 | 233099 | 82.44 | 82.35 | 82.24 | 233102 | 63.92 | 62.21 | 54.10 |
| 233103 | 86.62 | 85.90 | 82.33 | 233104 | 99.95 | 99.96 | 99.85 | 233106 | 68.11 | 68.81 | 55.45 |
| 233107 | 96.83 | 96.67 | 96.59 | 233108 | 95.83 | 95.83 | 95.83 | 233109 | 67.62 | 67.32 | 68.20 |
| 233110 | 85.99 | 86.35 | 86.06 | 233112 | 80.07 | 79.57 | 77.49 | 233113 | 99.95 | 97.95 | 85.34 |
| 233114 | 73.55 | 73.25 | 72.06 | 233115 | 87.09 | 88.11 | 87.60 | 233116 | 99.59 | 100.00 | 98.20 |
| 233117 | 93.72 | 93.94 | 90.69 | 233118 | 91.95 | 91.83 | 91.59 | 233119 | 97.47 | 92.66 | 85.53 |
| 233120 | 52.67 | 52.49 | 51.70 | 233121 | 98.37 | 98.41 | 96.93 | 233122 | 27.70 | 28.82 | 28.09 |
| 233123 | 65.29 | 65.13 | 62.11 | 233124 | 71.67 | 70.87 | 66.06 | 233126 | 94.02 | 88.13 | 93.16 |
| 233130 | 92.53 | 96.24 | 95.89 | 233131 | 74.26 | 71.86 | 74.63 | 233132 | 99.05 | 98.95 | 98.55 |
| 233133 | 69.18 | 68.75 | 69.03 | 233134 | 79.22 | 78.21 | 77.71 | 233135 | 94.05 | 94.43 | 93.95 |
| 233137 | 94.01 | 94.33 | 92.43 | 233142 | 97.17 | 97.06 | 96.06 | | | | |
| 233146 | 64.28 | 64.53 | 63.28 | 233143 | 92.53 | 92.13 | 92.59 | | | | |

Table 5: **Top-5 baselines.** The test set performance for the Regularization Cocktail against the Top-5 Most Frequent (Top-5 F) and the Top-5 Highest Ranks (Top-5 R) baselines.

At Table 6 we provide the results of all our experiments for the baseline, the individual regularization methods, and the regularization cocktail. All the results are calculated on the test set after retraining on the best-found hyperparameter configurations. The evaluation metric used for the performance is the balanced accuracy.

| Task Id | PN | BN | LA | SE | SWA | SC | AT | SS | SD | MU | CO | CM | WD | DO | Cocktail |
|---|---|---|---|---|---|---|---|---|---|---|---|---|---|---|---|
| 233090 | 84.13 | 86.78 | 83.99 | 86.48 | 87.96 | 87.21 | 86.92 | 84.28 | 87.21 | 89.27 | 85.60 | 86.77 | 87.06 | 86.92 | 89.27 |
| 233091 | 99.70 | 99.85 | 99.70 | 99.70 | 99.55 | 100.00 | 99.85 | 99.85 | 99.69 | 99.85 | 99.55 | 99.85 | 99.85 | 99.85 | 99.85 |
| 233092 | 37.99 | 41.91 | 36.14 | 37.31 | 25.94 | 53.42 | 38.79 | 55.61 | 53.26 | 42.19 | 32.48 | 42.22 | 35.76 | 38.70 | 61.46 |
| 233093 | 97.75 | 98.50 | 96.00 | 97.75 | 69.25 | 98.25 | 97.25 | 97.25 | 98.25 | 98.00 | 98.00 | 97.75 | 98.00 | 98.00 | 98.00 |
| 233088 | 69.40 | 68.69 | 70.83 | 69.76 | 69.40 | 66.43 | 69.29 | 66.43 | 67.14 | 70.00 | 70.36 | 64.29 | 69.29 | 68.10 | 74.64 |
| 233094 | 83.77 | 83.17 | 84.36 | 84.39 | 83.36 | 80.82 | 83.17 | 83.20 | 81.98 | 83.77 | 81.47 | 78.65 | 83.20 | 82.60 | 82.58 |
| 233096 | 70.27 | 66.56 | 71.95 | 76.43 | 75.44 | 77.40 | 71.95 | 65.31 | 78.31 | 72.43 | 76.84 | 74.94 | 67.33 | 72.98 | 74.38 |
| 233099 | 76.89 | 77.92 | 75.95 | 78.23 | 76.38 | 78.38 | 76.75 | 75.56 | 78.61 | 78.67 | 82.56 | 82.23 | 76.99 | 78.52 | 82.44 |
| 233102 | 61.00 | 62.89 | 61.32 | 63.57 | 56.67 | 60.79 | 59.99 | 43.04 | 60.77 | 61.95 | 63.30 | 63.49 | 64.03 | 63.75 | 63.92 |
| 233103 | 87.51 | 87.02 | 88.25 | 87.03 | 87.22 | 85.90 | 87.99 | 87.64 | 85.90 | 87.12 | 87.26 | 86.59 | 86.74 | 88.39 | 86.62 |
| 233104 | 99.97 | 99.96 | 99.96 | 99.94 | 2.57 | 99.97 | 99.95 | 92.77 | 99.97 | 99.95 | 99.96 | 99.97 | 99.96 | 99.96 | 99.95 |
| 233106 | 62.83 | 68.90 | 62.46 | 65.70 | 62.16 | 61.85 | 61.89 | 44.63 | 62.05 | 66.29 | 65.43 | 64.99 | 66.50 | 67.04 | 68.11 |
| 233107 | 95.92 | 95.93 | 96.01 | 96.36 | 95.23 | 95.76 | 95.77 | 95.37 | 96.22 | 96.52 | 96.10 | 96.55 | 95.98 | 96.23 | 96.83 |
| 233108 | 87.50 | 91.20 | 85.65 | 87.96 | 50.00 | 93.98 | 92.59 | 94.91 | 94.44 | 94.44 | 93.06 | 95.37 | 91.67 | 90.74 | 95.83 |
| 233109 | 67.84 | 73.68 | 66.52 | 68.20 | 66.45 | 65.20 | 66.89 | 66.74 | 67.03 | 68.64 | 67.32 | 70.18 | 66.23 | 68.42 | 67.62 |
| 233110 | 78.08 | 72.58 | 72.70 | 83.40 | 66.93 | 72.74 | 74.12 | 70.16 | 74.76 | 74.09 | 85.71 | 85.76 | 72.34 | 83.14 | 85.99 |
| 233112 | 73.63 | 74.68 | 73.37 | 74.33 | 77.36 | 73.86 | 72.91 | 72.06 | 74.35 | 72.08 | 76.23 | 75.74 | 72.48 | 76.35 | 80.07 |
| 233113 | 99.47 | 99.89 | 99.92 | 99.87 | 55.86 | 98.11 | 99.46 | 90.60 | 98.11 | 99.94 | 99.92 | 99.91 | 99.88 | 99.89 | 99.95 |
| 233114 | 67.75 | 68.90 | 68.81 | 69.11 | 67.36 | 68.08 | 67.44 | 67.70 | 68.56 | 68.59 | 71.93 | 73.13 | 67.80 | 66.87 | 73.55 |
| 233115 | 86.27 | 85.79 | 88.73 | 86.44 | 87.26 | 87.74 | 88.39 | 87.74 | 88.39 | 88.73 | 88.25 | 88.90 | 87.91 | 86.27 | 87.09 |
| 233116 | 97.44 | 100.00 | 96.79 | 97.44 | 87.35 | 99.47 | 99.14 | 97.46 | 99.69 | 99.37 | 97.64 | 99.04 | 97.44 | 99.69 | 99.59 |
| 233117 | 94.81 | 92.86 | 93.51 | 93.51 | 90.48 | 93.72 | 92.86 | 92.64 | 93.72 | 93.51 | 93.07 | 93.72 | 93.94 | 94.59 | 93.72 |
| 233118 | 90.46 | 90.86 | 90.73 | 90.75 | 81.72 | 89.91 | 90.69 | 86.69 | 90.06 | 91.11 | 91.09 | 91.88 | 90.70 | 90.51 | 91.95 |
| 233119 | 97.06 | 93.76 | 97.79 | 96.08 | 92.15 | 87.83 | 97.16 | 87.08 | 87.68 | 98.14 | 96.50 | 97.51 | 97.33 | 97.24 | 97.47 |
| 233120 | 50.26 | 50.95 | 51.29 | 50.50 | 51.63 | 50.92 | 50.23 | 51.00 | 50.23 | 50.72 | 52.35 | 52.10 | 50.41 | 50.30 | 52.67 |
| 233121 | 96.12 | 97.83 | 96.45 | 96.74 | 92.40 | 95.31 | 96.34 | 91.38 | 95.15 | 97.52 | 97.88 | 97.80 | 96.88 | 97.00 | 98.37 |
| 233122 | 16.84 | 22.26 | 17.20 | 19.65 | 20.90 | 24.53 | 16.77 | 18.71 | 24.35 | 23.62 | 23.43 | 24.10 | 17.52 | 23.98 | 27.70 |
| 233123 | 51.51 | 51.74 | 50.86 | 53.16 | 56.11 | 53.58 | 49.65 | 49.88 | 51.74 | 51.22 | 60.98 | 61.67 | 51.13 | 55.12 | 65.29 |
| 233124 | 65.08 | 66.82 | 65.57 | 66.56 | 66.15 | 57.71 | 65.26 | 64.97 | 58.04 | 67.24 | 70.03 | 68.84 | 66.86 | 67.00 | 71.67 |
| 233126 | 90.64 | 58.17 | 90.42 | 92.94 | 92.60 | 93.99 | 90.45 | 88.55 | 93.98 | 93.58 | 93.86 | 93.87 | 92.97 | 94.10 | 94.02 |
| 233130 | 87.76 | 87.81 | 88.98 | 88.99 | 70.72 | 87.99 | 50.00 | 85.25 | 88.35 | 92.43 | 50.00 | 95.81 | 94.92 | 91.19 | 92.53 |
| 233131 | 70.94 | 69.28 | 71.59 | 70.94 | 71.31 | 72.14 | 71.59 | 71.59 | 72.32 | 70.94 | 72.69 | 72.42 | 70.76 | 70.76 | 74.26 |
| 233132 | 96.93 | 98.62 | 97.52 | 97.14 | 94.58 | 96.85 | 97.00 | 97.27 | 96.90 | 98.66 | 98.14 | 99.15 | 96.81 | 96.73 | 99.05 |
| 233133 | 63.71 | 65.11 | 65.00 | 66.05 | 64.57 | 66.21 | 62.82 | 64.33 | 65.98 | 68.75 | 66.58 | 66.28 | 64.36 | 64.81 | 69.18 |
| 233134 | 78.05 | 75.87 | 79.05 | 78.22 | 80.38 | 78.38 | 76.88 | 78.38 | 78.38 | 76.88 | 77.38 | 76.54 | 76.88 | 76.21 | 79.22 |
| 233135 | 93.07 | 92.49 | 92.10 | 93.17 | 93.17 | 92.10 | 93.17 | 93.27 | 92.10 | 92.58 | 92.68 | 94.53 | 93.75 | 93.36 | 94.05 |
| 233137 | 91.91 | 93.71 | 92.16 | 92.56 | 90.38 | 91.58 | 91.36 | 88.09 | 91.60 | 92.72 | 92.48 | 92.39 | 92.95 | 92.72 | 94.01 |
| 233142 | 92.33 | 96.70 | 92.90 | 92.35 | 63.59 | 95.47 | 94.37 | 93.60 | 95.56 | 95.56 | 93.81 | 93.25 | 92.60 | 93.85 | 97.17 |
| 233143 | 50.00 | 92.30 | 92.76 | 50.00 | 70.81 | 90.28 | 50.00 | 50.31 | 89.26 | 50.00 | 50.00 | 50.00 | 92.26 | 50.00 | 92.53 |
| 233146 | 63.12 | 60.06 | 62.79 | 64.16 | 63.39 | 64.42 | 63.52 | 54.64 | 64.21 | 64.26 | 64.05 | 64.57 | 64.41 | 64.37 | 64.28 |

Table 6: **Detailed Table of Results.** The test set performance for the plain network, individual regularization methods and for the regularization cocktails.

Lastly, in Table 7 we present the results of Experiment 3 where we compare our method with GBDT. The results describe the balanced accuracy calculated on the test set after retraining on both methods on the best hyperparameter configuration found within the given budget.

| Task Id | GBDT | Cockt. | Task Id | GBDT | Cockt. | Task Id | GBDT | Cockt. | Task Id | GBDT | Cockt. |
|---|---|---|---|---|---|---|---|---|---|---|---|
| 233090 | 90.000 | 89.270 | 233091 | 99.850 | 99.850 | 233092 | 46.850 | 61.461 | 233093 | 97.500 | 98.000 |
| 233088 | 71.191 | 74.643 | 233094 | 80.165 | 82.576 | 233096 | 63.353 | 74.381 | 233099 | 79.830 | 82.443 |
| 233102 | 62.764 | 63.923 | 233103 | 88.341 | 86.619 | 233104 | 99.967 | 99.953 | 233106 | 68.947 | 68.107 |
| 233107 | 97.217 | 96.826 | 233108 | 93.519 | 95.833 | 233109 | 64.985 | 67.617 | 233110 | 72.283 | 85.993 |
| 233112 | 72.645 | 80.073 | 233113 | 98.571 | 99.948 | 233114 | 72.926 | 73.546 | 233115 | 88.589 | 87.088 |
| 233116 | 100.000 | 99.587 | 233117 | 93.074 | 93.723 | 233118 | 90.457 | 91.950 | 233119 | 83.070 | 97.471 |
| 233120 | 52.421 | 52.668 | 233121 | 77.897 | 98.370 | 233122 | 21.144 | 27.70 | 233123 | 55.593 | 65.287 |
| 233124 | 63.428 | 71.667 | 233126 | 94.137 | 94.015 | 233130 | 91.797 | 92.535 | 233131 | 74.447 | 74.262 |
| 233132 | 98.704 | 99.049 | 233133 | 70.120 | 69.183 | 233134 | 78.878 | 79.217 | 233135 | 95.119 | 94.045 |
| 233137 | 74.620 | 94.01 | 233142 | 13.534 | 97.17 | 233143 | 92.514 | 92.531 | 233146 | 58.201 | 64.280 |

Table 7: **Results of Experiment 3** The performances of the Regularization Cocktail and the GBDT algorithm over the different datasets.

