# OpenReview forum: "Regularization Cocktails for Tabular Datasets"
_ICLR.cc/2021/Conference — Reject_

### Official Review · AnonReviewer2 · 2020-10-27
**No contribution?**

**Rating:** 6
**Confidence:** 4

**Review:**

The paper presents a study on regularization methods for the feedforward fully connected neural networks.
The study is formulated as hyper-parameter optimization task, heavily using Auto-Pytorch library. The paper claims as contribution (sorry for copy-pasting):

1. We demonstrate the empirical accuracy gains of regularization cocktails in a systematic
manner via a large-scale experimental study;
2. We challenge the status-quo practices of designing universal dataset-agnostic regularizers,
by showing that an optimal regularization cocktail is highly dataset-dependent;
3. We demonstrate that regularization cocktails achieve a higher gain on smaller datasets;
4. As an overarching contribution, this paper provides previously-lacking in-depth empiri-
cal evidence to better understand the importance of combining different mechanisms for
regularization, one of the most fundamental concepts in machine learning.

***

I am highly sceptical on the paper usefulness for the community.
In general terms, the benchmark/empirical study type of paper typically can have one (or more) of the following contributions:

a) New knowledge, which was obtained as a result of a study. E.g. surprising results, practical recommendations, so on. For example, A Metric Learning Reality Check by Musgrave et. al (ECCV 2020) revealed surprising knowledge about metric learning methods.

b) The methodology of such study, which was not used before. E.g. Visual Object Tracking Challenge, which become the benchmark for the tracking methods since 2013.
c) The software and/or dataset, which were developed for the study. E.g. OpenAI Gym.


(a) is not the case IMO, because all the recommendations are known to the practicioners, e.g. check the any Kaggle winning solution
https://www.kaggle.com/sudalairajkumar/winning-solutions-of-kaggle-competitions

(b) I see no novelty in using hyper-parameter optimization for the study. The paper agrees with me on this  (see Related Work, "Positioning in the realm of AutoML"

(c) Neither software, nor dataset is proposed -- the paper uses existing ones.

****

Now I will go over contributions.


1. It is well known that the regularization/augmentation/... need to be tuned to archieve the best results. One can publish a CVPR paper about such good combination, e.g. He et.al. (CVPR2019),
Bag of Tricks for Image Classification with Convolutional Neural Networks

2. I don't see the support for that claim in the paper. Yes, the specific combination of regularization techniques, which performs the best on the given dataset is, perhaps, unique. But the techniques are applicable broadly, which is supported by the paper (Fig. 1), e.g. DropOut, MixUp, BatchNorm, are pretty universal.

3. It is also obvious, that the less data you have, more regularization and design bias are needed for better results, see OpenAI Image GPT, or more ViT paper vs. ConvNets.

4. See (1)


Overall, if the paper spend some space on particually interesting regularization combinations/interplay of components/analisys, it might be quite useful for researchers. For now it seems as a lot of experiments were done, but no analisys is really performed.

E.g. abstract says: "there is no systematic study as to how different regularizers should be combined into the best "cocktail"".

But I don't see the answer of how should they be combined.

********

Small things, not contributing to the score:

- AutoPyTorch is cited twice, as arXiv and as CoRR

- While skip-connections and BN can be seen as "regularization", I would rather call them "architecture". Anyway, that is just matter of naming.


### After rebuttal update.

The paper has been significantly refocused and now sells itself as a way of deep neural networks being competitive versus gradient boosting methods, which are dominating the tabular heterogeneous tasks.

I was also convinced by authors responce on paper novelty, technical contribution and (after the re-focusing) potential usefulness to the community.
Thus, I am raising my rating to weak accept.

### Comments on authors response (as after Nov 24 I cannot post messages, visible to authors)

> 1. However, we would be thankful if you might share any prior work (paper or published practice) where the authors automatically searched for the optimal combination of regularizers for deep learning models among a large set of regularizers, as presented in this study.

I am surprised with the results of the googling, but have to admit that authors are technically right and I was wrong. While it seems obvious to me, that regularization (specifically, dropout and L2 weight decay) are the hyperparameters of the deep network training, somehow papers and guides online mostly consider mostly architectural things + learning rate + (sometimes) dropout rate as a hyperparameters to optimize.

https://arxiv.org/pdf/2006.12703.pdf
https://nanonets.com/blog/hyperparameter-optimization/

Anyway, I lift my objections on novelty.

> 3. "Neither software, nor dataset is proposed, the paper uses existing ones." : We engineered a source code that selects the application of 13 regularizers to a neural network, which required extensive programming efforts and several additions to the AutoPytorch library, as mentioned in Section 4.1.

OK, I agree.


> 4. "It is well known that the regularization/augmentation/ methods need to be tuned ... He et.al. (CVPR2019)" : The suggested reference is a collection of refinements (many of which are actually not regularization techniques), which have been suggested by the deep learning community for maximizing the generalization on Imagenet. That paper only summarizes a collection of some practices, however, it does not present a method that searches for the best combination of a large set of practices

No, we are discussing different things. I gave the He at.al as an example, that community is well aware about the fact, the regularization and augmentation (and other things) have to be highly tuned. I agree that He at.al and the current paper use completely different methods for solving the problem (manual tinkering vs. auto-search). What I disagree is that the community was not aware about importance of regularization tuning before this paper.

>5.  E.g. DropOut was present in only 35% of the dataset cocktails, hence was not selected in the cocktails of the 65% of the datasets.

And experiments were using fixed-size network. Of course, is the network is not wide enough for the task, the dropout might not be needed. It is also quite strong statement that "there is no universal regularization", given that L2 weight decay and dropout are widely used in a such different domains as image, text, speech processings, RL and so on.
****

I would like to point out the TabNet paper https://arxiv.org/pdf/1908.07442.pdf, which claimed "beating GB methods for the tabular data". I appreciate the fact, that unlike the TabNet, RegCocktails were using a standard deep MLP and not the attention model, yet one needs to add that reference.

---

> ### Author Response · Authors · 2020-11-22
> **Response to AnonReviewer2**
>
> 1) **"... the recommendations are known to the practitioners, e.g. check any Kaggle winning solution https://www.kaggle.com/sudalairajkumar/winning-solutions-of-kaggle-competitions"** : In our search we found no previous paper that tackles the same problem: discovering the best combination of regularizers in an automatic manner from a large pool of regularizers. However, we would be thankful if you might share any prior work (paper or published practice) where the authors automatically searched for the optimal combination of regularizers for deep learning models among a large set of regularizers, as presented in this study. We believe that apart from being the first paper that learns optimal regularization cocktails, we also provide ample evidence on the efficiency of our approach over a large-scale benchmark that features tabular datasets from different domains.
> 2) **"... no novelty in using hyperparameter optimization for the study"** : The novelty is in rethinking the optimal fusion of different regularization methods as an instance of a HPO problem and demonstrating its empirical superiority against existing regularizers. In that sense, the actual HPO solving algorithm (BOHB in our case) is just a tool for solving our problem, in a similar way to e.g. using Adam+SGD as an optimization tool to train a new architecture, or a new loss function. However, we emphasize that the problem formulation is new, the overall solution is applied for the first time on the problem, and the empirical findings are strong and significant.
> 3) **"Neither software, nor dataset is proposed, the paper uses existing ones."** : We engineered a source code that selects the application of 13 regularizers to a neural network, which required extensive programming efforts and several additions to the AutoPytorch library, as mentioned in Section 4.1.
> 4) **"It is well known that the regularization/augmentation/ methods need to be tuned ... He et.al. (CVPR2019)"** : The suggested reference is a collection of refinements (many of which are actually not regularization techniques), which have been suggested by the deep learning community for maximizing the generalization on Imagenet. That paper only summarizes a collection of some practices, however, it does not present a method that searches for the best combination of a large set of practices. In contrast, our method proposes an approach to automatically discover the optimal subset of regularization techniques for a specific dataset from a large set of available regularizers for neural networks. Therefore, we believe that our paper presents a novel approach in automatizing the combination of regularizers that to the best of awareness has never been tried and analyzed before. If the reviewer is aware of any previous paper that proposes a close idea, we would be glad to know.
> 5) **"... DropOut, MixUp, BatchNorm, are pretty universal."** : We found no “universal” regularization methods when used in combination as Experiment 2 in Section 5 shows. E.g. DropOut was present in only 35% of the dataset cocktails, hence was not selected in the cocktails of the 65% of the datasets. In fact, searching for optimal combinations of subsets of regularization techniques that are dataset-specific is a novel problem that this paper addresses for the first time.
> 6) **"...the less data you have, more regularization and bias is needed"** : You are right on this point. Even though the experiment is correct and the outcome clear, still the result can be perceived as “obvious”. We agree with you (and other reviewers) and decided to drop this experiment to make room for the new results comparing the cocktails against Gradient Boosted Decision Trees (GBDT).
> 7) **"there is no systematic study as to how different regularizers should be combined"** : Thanks for your comment and suggestions which we appreciate. Our answer to the question “How to combine regularizers?”, is that the combination of regularizers should be treated as a Combined Algorithm Selection and Hyper-parameter (CASH) problem and solved specifically on each dataset. Unfortunately, because every optimal combination is dataset-dependent our experimental findings suggest there is no “magic recipe” that is optimal on all datasets. We actually did provide an analysis of the most frequent regularizers within the best combinations (cocktail) of each dataset, as well as the frequent pairs of regularizers. Furthermore we compared against top-5 universal cocktails of strong, or frequent regularizers and found them to be sub-optimal w.r.t. our approach (pls notice the new experiments). The frequencies of cocktail ingredients show that there is no pair of methods whose interaction is optimal in more than 50% of the datasets (please see Figure 3 of the updated draft). So general recipes in the form of “always use method A together with method B” are unfortunately not evidenced by our large-scale experiment, as optimal combinations are highly  dataset-dependent.

---

> > ### Comment · AnonReviewer2 · 2020-11-22
> > **Quick response to the general idea of the comment**
> >
> > I appreciate the detailed answer and the paper revision, which I need to evaluate in more detail and a bit more time for it.
> >
> > However, I would like to point out one misunderstanding between author's position and my position as a practitioner.
> > I agree, that "one should treat finding regularization as HyperOpt problem, and NOT solve it with Random/Grid search" which paper proposes, can be seen as a novel idea.  It is also kind of in agreement with Sutton'`s "bitter lesson".
> >
> > However, I, as a practitioner, don't really like the idea "in order to get good performance on the single dataset, you always should run an expensive HyperOpt search".
> > I mean, yes, that is probably true, but that is quite expensive, to say the least.
> >
> > It also looks like that I am a wrong person to review AutoML papers because of the such mindset.

---

> > > ### Author Response · Authors · 2020-11-22
> > > **Response to AnonReviewer2**
> > >
> > > Dear AnonReviewer2,
> > >
> > > You are raising an important point, we fully agree that the AutoML-inspired HPO of our regularization cocktail is time-consuming in terms of the runtime budget (especially for very large datasets). However, the manual fine-tuning and calibration of methods to fit a new dataset is also time-demanding. As models do not “plug-and-play” (i.e. generalize directly) on new datasets, practitioners need to revise the inductive biases of their model choices continuously until a generalized version is achieved. In the end of the day, complex HPO methods demand a large computational time, however the manual design of efficient models takes both a long time and considerable manual labor.
> > >
> > > On the other hand, the reviewer has a good point if we would be thinking about the typical deep learning of very large, raw data (e.g., training on ImageNet), where such a full HPO from scratch is indeed ludicrously expensive. But (1) on tabular data (focus of this paper) this tuning is perfectly reasonable, and (2) knowing that the regularization cocktails perform so well will almost certainly trigger a lot of follow-up work aiming to reduce the computational overhead of finding the best cocktail for a specific dataset.
> > >
> > > Best,
> > > The Authors

---

### Official Review · AnonReviewer3 · 2020-10-28
**Review of "Regularization Cocktails"**

**Rating:** 6
**Confidence:** 5

**Review:**

Summary:
This paper provides an empirical study of combining different regularizers. Fourteen regularizers including batch norm, weight decay, etc. are considered. The authors use BOHB (Falkner et al. 2018) to optimize for whether each regularizer is active, and additional regularizer-specific hyperparameters. Using 40 tabular datasets, they show mixtures nearly always outperform tuning a single regularizer, and that the benefits of regularization improves for smaller datasets.

Strengths:
- To my knowledge, this is the first paper that does an empirical study of combining regularizers.
- The list of regularizers considered are extensive.
- The paper is very well written and easy to follow. The figures are nice and easy to understand.

Weaknesses:
I think the biggest weakness of this work is that it is not very useful practically.
- All experiments are run on tabular data. I think it’s fair to say that most practitioners who deal with regularization are interested in non tabular data, given the fact that the regularization methods were individually developed for training on non-tabular data. I find it hard to imagine extrapolating results on tabular data to images, or text.
- The paper frames itself as a “methods” paper more than an analysis paper, where the main claims revolve around the superiority of the regularization cocktail. In fact, the take-home messages given in the conclusion explicitly recommends the use of the regularization cocktail. I find this advice not very useful because: 1) the experiments were run on tabular data, 2) regularization cocktail is too expensive to justify the improvement (if any).
- The conclusions are trivial. It’s quite obvious that a more general method always does at least as good as the method it subsumes, as long as the tuning of the parameters can be done sufficiently. The fact that the regularization cocktail does better than individual methods, or a combination of a few, is very believable in the tabular setting, since the tuning can be done sufficiently. The fact that more regularization is needed for smaller datasets is also well-known.

The paper would be more useful with a similar experimental protocol applied to a non-tabular dataset (CIFAR-10, Fashion MNIST, SVHN, etc), but with a focus on analyzing trends (rather than highlighting the mixture method), or comparing with SOTA human designed cocktails. I recommend reject because of the lack of practicality of the paper.

Comments:
- For the experiments corresponding to Figure 5, I wonder what the gap would be like for the best performing individual regularizer. I am curious because I think for a small dataset, the need for a state of the art regularizer diminishes, and just one good regularizer suffices.

=== update ===

I have read the revised paper, and decided to update the score (see response to authors' comment).

---

> ### Author Response · Authors · 2020-11-22
> **Response to AnonReviewer3**
>
> 1) **"All experiments are run on tabular data. I think it’s fair to say that most practitioners who deal with regularization are interested in non tabular data ... I find it hard to imagine extrapolating results on tabular data to images, or text."** : It is true that a large section of the deep learning community focuses on computer vision, but there are plenty of application domains that have tabular data (medical domain, advertisement business, recommender systems, etc.). Furthermore, regularizing neural networks for tabular data is quite crucial because deep learning is not the state-of-the-art on tabular data and is outperformed by Gradient Boosted Decision Trees (GBDT). After carefully thinking about your point we decided that the best way to prove our point, is to show that by regularizing neural networks carefully we can achieve state-of-the-art and outperform GBDT. We ran extensive new results against GBDT with per-dataset tuned hyper-parameters and we are presenting them in Section 5 of the new paper draft. The findings show that the regularization cocktail outperforms Gradient Boosting with a significant margin. This is the very first time in our knowledge that neural networks outperform GBDT. In clear contrast to the belief of the community, the new experiments demonstrate that principally-regularized deep learning models can achieve state of the art in tabular data via a large-scale experimental protocol, which can have a game-changing impact on the perception that “deep learning doesn’t work for tabular data”.
> 2) **"The paper frames itself as a “methods” paper more than an analysis paper, where the main claims revolve around the superiority of the regularization cocktail. In fact, the take-home messages given in the conclusion explicitly recommends the use of the regularization cocktail. I find this advice not very useful because: 1) the experiments were run on tabular data, 2) regularization cocktail is too expensive to justify the improvement (if any)."** : Please find our response to the usage of tabular data above. Regarding the statement that the cocktail is too expensive to justify the improvement, we believe it does not correctly summarize the findings. On one hand the gain is statistically significant and the improvement against baselines is very considerable by any existing prior practice of comparing machine learning models. On the other hand, the HPO runs we presented were completed within a few days on CPU servers for large datasets with up to half a million instances.
> 3) **"The conclusions are trivial. It’s quite obvious that a more general method always does at least as good as the method it subsumes, as long as the tuning of the parameters can be done sufficiently. The fact that the regularization cocktail does better than individual methods, or a combination of a few, is very believable in the tabular setting, since the tuning can be done sufficiently. The fact that more regularization is needed for smaller datasets is also well-known."** : To the best of our awareness there is no prior work which demonstrates that a per-dataset learned collection of regularizers outperforms common regularizers. Neither is the usage of per-dataset regularization cocktails a common practice by any prior paper we are aware of, nor it is a common practice by practitioners. We are actually happy that the reviewer found the outcome of our work “believable” and we would be glad to address any further technical criticism of our work.
> 4) **"The paper would be more useful with a similar experimental protocol applied to a non-tabular dataset ... but with a focus on analyzing trends (rather than highlighting the mixture method), or comparing with SOTA human designed cocktails."** : In addition to the points we responded above, we would like to further address the concern on the lack of practicality with the new evidence on the comparison against the state-of-the-art in tabular datasets. We demonstrate in Section 5 that the regularization cocktail makes neural networks outperform Gradient Boosting in tabular data, which to the best of our knowledge happens for the first time and changes the community perception that “deep learning does not work on tabular data”. If the reviewer is happy with the rebuttal, but s/he conditions his change of opinion on making it clear that the paper is focused on tabular data, then we offer to change the title to “Regularization Cocktails on Tabular Datasets”.
> 5) **"For the experiments corresponding to Figure 5, I wonder what the gap would be like for the best performing individual regularizer. I am curious because I think for a small dataset, the need for a state of the art regularizer diminishes, and just one good regularizer suffices."** : We agree that although this experiment is technically correct, it does not show a novel finding (the other reviewers agree with you). So we decided to drop it from the paper and make room for presenting the comparison to GBDT.

---

> > ### Comment · AnonReviewer3 · 2020-11-24
> > **I have updated my score.**
> >
> > I have read the other reviewers’ comments, the authors’ responses, and the updated paper. Given that the paper now seems to be focused on tabular datasets, Most of my concerns are addressed, and therefore, I’m increasing the score to weak acceptance. I would like the abstract, introduction, and section 4.1 to be further changed to make the focus on tabular dataset more clear.

---

> > > ### Author Response · Authors · 2020-11-25
> > > **Strengthening emphasis on Tabular Datasets**
> > >
> > > Dear AnonReviewer3,
> > >
> > > We strengthened the emphasis on tabular datasets following your recommendation. Tabular datasets are mentioned multiple times now in the abstract, introduction (incl. a motivation why we focus on tabular data), and experimental protocol. We also changed the title accordingly.

---

### Official Review · AnonReviewer1 · 2020-10-28
**Interesting study, requires further analysis**

**Rating:** 6
**Confidence:** 4

**Review:**

Summary: This work takes a step towards understanding the effect of automated selection of regularisation techniques and analyses the results across 42 structured datasets. It defines a search space over 13 regularisation techniques and employs one flavour of Bayesian Optimisation + Hyperband approach to find an optimal combination of regularisers. It concludes by substantiating three claims with corresponding experiments.

The work addresses an interesting problem that is relevant for full-scale automation of deep learning. The paper is easy to follow with a thorough literature survey (to my knowledge) and contains relevant experiments.

I find the study can improve by addressing the following
- Firstly, it is difficult to argue for generalisability of this work as currently this work relies on only one algorithm to find the optimal combination of regulaisers. One would need to check how much influence does BOHO have on the claims made within this work.
- The three hypotheses are insightful but limited to small scale datasets. In particular, the third claims about gains on small datasets stands out as not-so-surprsing but challenges the effectiveness of the cocktail-system as DL models often find use in large scale problems.
- While the study provides some descriptive insights, it falls short on prescriptive solutions for automated selection of regularisers.
- It is interesting to note that in Exp 1 CutMix, CutOut and Mixup rank better as individual techniques and yet only CutMix crosses the threshold of 0.3 for Figure 3. Did the authors find any explanation for this? It would be interesting to see how Figure 4 changes if these are included along with BN, DO and WD.
- Finally, I think the paper can improve with some additional information and more discussion. For instance, a brief description of BOHO, details on what the scatter plot correspond to in Figure 4, insights and discussion of results from all figures in the supplementary etc. It would also be important to establish statistical significance of the experiments with multiple re-runs.

---

> ### Author Response · Authors · 2020-11-22
> **Response to AnonReviewer1**
>
> 1) **"Firstly, it is difficult to argue for generalisability of this work ... would need to check how much influence does BOHO have on the claims made within this work."** : BOHB is the state-of-the-art algorithm in AutoML for tuning hyper-parameters of gray-box models such as neural networks (in graybox optimization we make usage of intermediate performances, e.g. validation error on a small number of epochs). The method is a combination of Hyperband with surrogate models and is the natural choice for our setup. Furthermore, our setup is fair because all baselines were tuned by the same state-of-the-art HPO solver with ample hyper-parameter runtime on each dataset (up to 4 days per tabular dataset is quite sufficient) on multi-core CPUs (please see Section 4.1). However, we would like to emphasize that the quality of the discovered hyper-parameters is not actually very dependent on the choice of the HPO method if the runtime budget is large (as in our setup). Even random search will eventually perform as good as any other method given an abundant amount of trials. Considering that: 1) BOHB is the state-of-the-art and the natural choice for neural networks, 2) the same HPO algorithm was used for the baselines under the same conditions, and  3) that we gave each dataset a large budget for HPO, then, we do not see a bias in the selection of the HPO method.
> 2) **"... gains on small datasets stands out as not-so-surprsing but challenges the effectiveness of the cocktail-system as DL models often find use in large scale problems."** : After carefully reading the reviewers, which mostly stated that this experiment is not “wrong”, but neither it demonstrates a novel finding, we decided the paper would be much better if we delete it entirely and make room for our new experiment that compares the regularization cocktail against the state-of-the-art classifier Gradient-Boosted Decision Tree.
> 3) **"While the study provides some descriptive insights, it falls short on prescriptive solutions for automated selection of regularisers.""** : We take responsibility for not making this point crystal clear. Our finding is that there is no magic recipe on combining regularizers. We are aware that the community would love to hear findings such as “e.g. if you use method A, B then you should combine it with method C”, but in our results we found no clear pattern as it always depends on a dataset (see the cocktail frequencies of Section 5, Figure 3). It does not mean, e.g. that dropout is not a good regularizer, but that it does not always help when combined with other regularizers. We stress that the combination of regularizers which achieve the best validation accuracy is always dataset-dependent (It actually is very intuitional because of the different effects that interactions of regularizers have on each dataset, which demand a dataset-specific surrogate model for capturing such dataset-dependent hyper-parameter interactions). So our novel prescriptive solution is: Treat regularizing your model as a Combined Algorithm Selection and Hyper-parameter (CASH) problem and solve it for each new dataset. Please notice that the many existing prescriptive solutions of the deep learning community are largely a naive instance of our problem. They are mostly trial-and-error combinations of regularizers on ImageNet, so the community can be perceived to act as a distributed hyper-parameter optimization algorithm discovering regularization cocktails on ImageNet by human-run trials. At the end of the day, our work shows an accurate and automatized solution for replacing this inefficient process with a parametric HPO method.
> 4) **"CutMix, CutOut and Mixup rank better as individual techniques and yet only CutMix crosses the threshold of 0.3 for Figure 3. It would be interesting to see if these are included."** : The low frequencies arise from the fact that only one data augmentation method was permitted in a cocktail (as was specified in Section 3.2 and on Table 1 of the submitted version). But we admit that the point raised by the reviewer is a very good point here! To answer it, we invested quite some time to program an upgrade to our source code that allows the joint combination of multiple data augmentation techniques (hence the delay of the rebuttal). We are presenting two experiments as a follow-up to your suggestion: 1) repeated all the cocktail HPO and evaluation runs with the upgrade on multiple active data augmentation methods, and added two new baselines on the top-5 most frequent and stronger baselines (as you suggest BN, DO, WD, plus data augmentation). The results are presented in Sections 5, Figure 4 of the new draft.
> 5) **"... Finally, I think the paper can improve with some additional information and more discussion."** : Thank you for your feedback, we described BOHB in the Appendix, added details on what the scatter plot in Figure 4 corresponds to and added better descriptions on the supplementary materials

---

### Official Review · AnonReviewer4 · 2020-11-08
**A bag of experiments regarding parametric mixes of regularizers**

**Rating:** 6
**Confidence:** 4

**Review:**

*Summary and contributions:*
This paper provides empirical evidence showing that optimizing a parametric mix of regularizers when training a model provides better generalization than using handcrafted ones.
The results show that the best regularizers mix is dataset dependent, and that regularization matters most when limited data is available.


*Originality:*
This paper falls into the category “more parameters in the system lead to better results”.
I am not aware of a specific paper that has explored using a parametric mix of regularizer.
I suspect some previous papers might have explored smaller variants of this idea, but none with the systematic approach and breath of this paper.


*Significance:*
This paper is part of the AutoML trend. It provides support to the idea that the regularizers should also be thrown in the optimization target box.
(Together with network architecture, data augmentation, and the optimizer themselves).
As such, it provides recommendations for best practices in machine learning.


*Strengths:*
* The paper carefully explicits the hypotheses being tested, and design experiments to probe them.
* The observed effect is significant to the point of justifying the additional system complexity.
* The experiments include a reasonable amount of regularization techniques in the mix (and for comparison).


*Weaknesses:*
* The results are somewhat expected. More parameters make systems better, everything is task dependent, and that regularization matters more when less data is available.
* I did not find in the paper experiments adding the validation set in the training set (aka “what if we do not split ?”). Id est, a baseline accounting for the fact that in this setup the validation set is part of the overall training set.
* Experiments only consider tabular datasets for classification. The intuition indicates that the results would generalize to 1d audio, 2d image, or 3d point-cloud data (as well as for other tasks beyond classification); but it would be better to have some data points on this aspect.


*Clarity:*
The writing is clear, and the text is overall easy to follow.


*Correctness:*
I did not find anything particularly wrong.


*Relation to prior work:*
The related work section is satisfactory.
I am not aware of a specific related paper should be discussed (and a quick online search did not show either).


*Reproducibility:*
The general idea seems easily reproducible. The text provides enough detail to be able to reproduce the overall system.


*Specific feedback:*
- Section 3.1, footnote: hold-out validation versus cross-validation might not be a detail, since it affects what is considered the training set of the baseline.
It would be good to include results of the baselines with “default parameters” trained over train+val.
- Section 3.2: please mention the total number of additional parameters added to the system (or at least its order of magnitude).
- Section 4.1: please mention that the task considered is always classification, and the total number of datasets considered.
- Section 5, experiment 1: confirm our hypothesis that -> confirm our hypothesis 1 that
- Figure 3: what is the logic of the top row sorting ? Consider sorting by frequency of usage. Please put the numbers as percentages, to be consistent with the paper text.
- Section 5, experiment 3: just to confirm, the validation set size was kept fix ?
- Section 5, experiment 3: “indicate that strong regularization”; maybe I missed a nuance, but the presented results show the consequences of _tuned_ regularization. This is different from _strong_ regularization. Is there a mention of the weights used for regularization in this case tend to be higher than for other setups ? That would be expected, but I did not see it presented.
- Section 5, experiment 3: “regularization can help...better...than generally perceived”; please cite a paper (or blog post) taking that position. I learned in ML classes that regularization matters when data is small, thus to my knowledge these results simply fit the classical ML theory (Vapnik-era Statistical learning theory).
- Figure 4: please mention what the points represent (a dataset each ?). Also use scientific notation for the number of examples.


*Updates after reviews and authors feedback: *
The updates from the author are appreciated and make the arguments of the paper clearer.
After reading the other reviews and discussions, I keep my original score of "6: Marginally above acceptance threshold".

---

> ### Author Response · Authors · 2020-11-22
> **Response to AnonReviewer4**
>
> 1) **"I did not find in the paper experiments adding the validation set in the training set ... baseline accounting for the fact that the validation set is part of the overall training set."** : Our apologies that this was unclear. For all the baselines and our method we combine the training and validation set before fitting the final model with the best hyper-parameter configuration for measuring the test error, exactly as you suggest. In fact that was stated in Section 4.3 of the paper, however, to improve readability we modified Section 4.1 to reinforce the point.
> 2) **" Experiments only consider tabular datasets for classification. The intuition indicates that the results would generalize to 1d audio, 2d image, or 3d point-cloud data (as well as for other tasks beyond classification); but it would be better to have some data points on this aspect."** : We intentionally chose to work with tabular datasets for multiple reasons, apologies that this was not clear. First of all, tabular datasets represent a major and important dataset modality that is currently in possession by the industries. Secondly, neural networks do not perform strongly in tabular datasets and are outperformed compared to state-of-the-art models such as Gradient-Boosted Decision Trees (GBDT). Our belief is that neural networks are high variance models for tabular datasets and if they can be regularized more thoroughly, then they can achieve competitive results on tabular datasets. In order to demonstrate this point, we added an extensive comparison to GBDT which is arguably the best classifier on tabular datasets. We tuned the parameters of GBDT by giving the same large HPO time budget as our regularization cocktail. The results, which we are presenting in Section 5 show that a data-set specific regularization outperforms the carefully-tuned GBDT with a statistically, significant margin. This is the first method we are aware of that actually shows that Neural Networks can outperform GBDT in a very-large scaled experiment with an experimental protocol that includes extensive hyper-parameter tuning. As a result, we believe the findings are not only a demonstration of the power of regularization cocktails, but also a game-changing message to the community that deep learning can be state-of-the-art on tabular data if regularized carefully.
> 3) **"It would be good to include results of the baselines with “default parameters” trained over train+val."** :  The reported results already show the test performance when testing all methods after training the best found hyper-parameter on train + val combined. We actually did more than running baselines with the default parameters since the reported results include tuned hyper-parameters of all baselines in a principled HPO protocol with the search space of Table 1.
> 4) **"mention the total number of additional parameters added to the system"** : If the reviewer means the hyper-parameters of the regularization cocktail, it is the search space of Table 1 with a total of 18 hyper-parameters (we further clarified it in section 3.2) . If the reviewer means the added neural network params, it remains roughly the same (with the minor exception below) as the unregularized network, because the reg. cocktail does not add parameters to the neural network, it only regularizes the existing params differently: I.e. the cocktail adds a hyper-parametrization to the HPO, not to the prediction model itself.  (Exception: Shake-Shake doubles the params because it creates a parallel branch of layers).
> 5) **"mention that the task considered is always classification, and the total number of datasets."** : Thank you for the feedback, we added the clarification in Section 4.1.
> 6) **"Consider sorting by frequency. Please put the numbers as percentages ..."** : Thank you for your feedback, we changed it accordingly at the new Figure 3.
> 7) **"Experiment 3, was the validation set size kept fix?"** : The validation set was proportionally down-scaled like the training set in order to provide a correct setup of data sub-sampling (the folds were always stratified). However, for the results to be comparable, the test set was always fixed.
> 8) **"Tuned regularization is different from strong regularization"** : We agree that the usage of the term “strong” was misleading and it means exactly as the reviewer states: a dataset-specific subset of regularizers where the hyper-parameters are “tuned” jointly. So they don’t have to have high values, it depends on the combination, which is always dataset-specific.
> 9) **"...regularization matters when data is small, thus to my knowledge these results simply fit the classical ML theory."** : We agree that this experiment shows trivial findings and does not add novelty. After carefully considering your comment and the other reviewers we decided that the paper would be much stronger if we simply remove this experiment from the paper and make room for the comparison to GBDT.

---

### Comment · AnonReviewer2 · 2020-11-22
**Answers/rebuttal?**

Dear authors,

I see that you have updated the paper ("28 Sep 2020 (modified: 11 Nov 2020)").
Could you please summarize the changes and answer the questions in reviews?

Best, R2

---

> ### Author Response · Authors · 2020-11-22
> **Response to AnonReviewer2**
>
> Dear AnonReviewer2,
>
> Our apologies for the misunderstanding, the paper was not updated until this moment. We have provided the answers and the updated version of the paper accordingly. We look forward to your feedback.
>
> Best,
> The authors

---

### Author Response · Authors · 2020-11-22
**Reply to all reviewers about changes**

Thank you to all the reviewers for their important feedback and suggestions. We would like to apologize for our delay in replying that was caused by the extensive new experiments we had to run for the rebuttal. Although we will reply below to all individual reviews, we would additionally like to highlight the main changes for the rebuttal:

1. In order to consider the positive criticism of the reviewers that the paper lacks a practical benefit to the practitioners, we ran additional experiments to show that neural networks trained with the regularization cocktails achieve state-of-the-art predictive accuracy. Therefore, practitioners will be interested in using our method because it is the new state-of-the-art in classifying tabular datasets.
    * To demonstrate the point we compared against Gradient-Boosted Decision Trees (GBDT), which is the de-facto state-of-the-
    art classifier for tabular datasets and the practitioners’ main current tool in winning data science competitions involving tabular
    datasets (e.g. Kaggle). It is widely believed in the community that GBDT outperforms Neural Networks in tabular datasets.
    * We provide a fair comparison by tuning the hyper-parameters of GBDT on each dataset with the same train, val, test spits and
    the same hyper-parameter search time. The experiments suggest that the regularization cocktails are more accurate than GBDT
    with a statistically-significant margin. To the best of our knowledge this is the first paper that shows how to make neural
    networks outperform GBDT on tabular datasets, by rethinking the way neural networks are regularized.
    * The fact that regularization cocktails enable deep neural networks to outperform GBDT opens up the possibility for another
     huge area of applications of deep learning: small, tabular data sets.

2. Another great point of the reviewers was that we need to show a benefit not only against single regularization methods, but also against combinations of regularization techniques which are intuitive and universal for all datasets. To achieve the aim, we designed a new experiment 2, where we compare against:
    * A cocktail of the top 5 strongest regularization methods based on their performance on our dataset collection.
    * A cocktail of the top 5 most frequent regularization methods based on their occurrence frequencies on the dataset-specific
    cocktails.
    * The results show that these two cocktails of universal combinations of regularization methods, still underperform w.r.t. a
    dataset-specific optimal cocktails. It means that just combining best top recipes on regularizers does not achieve optimal
    results compared to the dedicated dataset-specific combinations.

3) The last major positive criticism of the reviewers was that the findings are somehow obvious. To address the concern we added new analysis on frequent combinations of regularizers and removed the plot that was showing the performance of  regularization on smaller dataset sizes.

4) We re-engineered our code and now stochastic weight averaging and snapshot ensembling can be both activated at the same time. We reran the experiments with this update and this combination shows a lift in several datasets. We modified Section 3.2 and we updated the figures and tables accordingly.

---

### Decision · Program_Chairs · 2021-01-07
**Final Decision**

**Decision:**

Reject

**Comment:**

The paper in its most recent version claims that deep neural networks, when very carefully regularized, outperform methods such as Gradient Boosting Trees on tabular data. This is genuinely surprising to me (in a good way), and I suppose it is as well to the community.

The paper initially received negative reviews with two key remarks that "The results are somewhat expected." (R4, R3, R2). Indeed, the original version mainly stated that very careful regularization helps on tabular data.  Naturally, the reviewers (including myself) seen then as the second key weakness that "All experiments are run on tabular data." (R4, R3).

Based on the reviews, the Authors have clarified and changed their message. I think it is well summarized by R2 "The paper has been significantly refocused and now sells itself as a way of deep neural networks being competitive versus gradient boosting methods, which are dominating the tabular heterogeneous tasks."

As R2 said and was reflected in comments by other reviewers, "[...] convinced by authors response on paper novelty, technical contribution and (after the re-focusing) potential usefulness to the community".

Given the new message of the paper, a key new question surfaces. Is this indeed the first convincing demonstration that deep learning can outperform more standard methods on tabular data? R2 pointed out TabNet (see also Google Cloud offering) that already in 2019 claimed "beating GB methods for the tabular data". There is also NeurIPS work "Regularization Learning Networks: Deep Learning
for Tabular Datasets"; their abstract opens with "Despite their impressive performance, Deep Neural Networks (DNNs) typically underperform Gradient Boosting Trees (GBTs) on many tabular-dataset learning tasks. We propose that applying a different regularization coefficient to each weight might boost the performance of DNNs by allowing them to make more use
of the more relevant inputs". The latter work did not claim to beat GBT. Regardless, the two works should be carefully discussed and compared empirically to in the new version of the work.

I am also not yet fully convinced by the added comparison to GDBT. Arguably, AutoML from the sklearn package is not the most popular way to use GDBT in practice. How would regularization cocktails compare to GDBT from XGBoost, optimized using either random search or bayesian optimization?

Based on the above, I have to recommend the rejection of the paper. The key reason is: *the new reframing of the paper is exciting but warrants a much more detailed and careful evaluation*.

I really appreciate the work the Authors have put in clarifying and changing the message of the paper. I understand this is disappointing that we won't be able to include the work in ICLR. Nevertheless, I hope that the Authors found the feedback useful, and wanted to thank the Authors for submitting the work for consideration in ICLR.